# Uncovering and Correcting Perception Model Weaknesses Using VLM-Based Analysis

## Abstract

This paper tackles the challenge of improving automated driving perception systems, focusing on rare, complex, or novel scenarios that previously deployed models fail to handle and developers struggle to identify. To address this, we propose a novel two-stage method (SPIDER ) to diagnose and resolve perception model insufficiencies using Vision Language Models (VLMs). In the first stage, we segment data in a semantic embedding space to identify regions containing visually similar samples that differ in detection performance. By comparing these high- and low-performance subsets, we use a VLM to extract semantic effects — interpretable factors correlated with model errors. In the second stage, these effects guide targeted data acquisition to improve the model. Samples representing the identified effects are selected, and the perception model is fine-tuned using this curated dataset. Evaluations on the NuScenes dataset demonstrate that SPIDER can effectively identify insufficiencies in the perception model and quantifies key parameters. SPIDER enhances model robustness and improves transparency and explainability, which are critical for safety in automated driving systems.

## 1 Introduction

In recent years perception systems have achieved remarkable capabilities with deep learning. At the same time, they remain extremely complex and opaque. Modern perception models behave as black boxes, making it difficult for developers to understand why the system fails in rare, complex, or shifting scenarios. (Li et al., 2024) This lack of transparency severely hampers debugging, as engineers have limited insight into the insufficiencies of a perception model and often cannot tell which visual factors (e.g., weather, lighting, object attributes, scene context) co-occur with insufficiencies. Thus, engineers must often resort to expert knowledge when trying to improve or fix errors (Kuznietsov et al., 2024). In safety-relevant applications like automated driving, such blind spots are unacceptable, as an unrecognized insufficiency can carry severe consequences.

Insufficiencies must often be identified within very large datasets, representing a broad Operational Domain (OD) with a multitude of potential influences on model performance. Without automated and interpretable analysis tools, manually scanning such data for subtle insufficiencies would be prohibitively time-consuming and practically infeasible. Therefore, we propose SPIDER as a solution which:

1. discovers and explains, at scale, which semantic factors correlate with an insufficiency within a given model.

2. leverages the explanations to guide improvement of the same model, with artifacts that can be referenced within a safety case.

We use VLMs to analyze the differences between high and low performing inputs to build an understanding of model's error modes. For example, while the model accurately detects trucks in general, SPIDER identifies that when view of the truck is obstructed by colored road barriers, the model struggles on trucks, but not on other vehicles, or on images with road barriers in general.

Beyond their role in model improvement, insufficiencies must also be traceable in the context of a safety argument. This requires not only detecting correlations but also providing explanations that are human-understandable.

Existing approaches only partially meet this need. Out-of-distribution detection and novelty discovery highlight unusual inputs but do not localize or characterize model-specific insufficiencies. Embedding-space clustering can find error-prone regions in data, yet typically answers where errors occur rather than why, and often requires strong supervision or extensive human-in-the-loop analysis. Cross-modal techniques and captioning methods provide descriptive fidelity but are not designed to extract causally suggestive contrasts between high- and low-performance samples, nor to close the loop to data acquisition and re-training. Consequently, detection and explanation are frequently decoupled, and explanation and remediation are rarely integrated into a repeatable pipeline.

We solve this by finding semantic regions with high performance differences and by employing a VLM to generate human understandable explanations for identified patterns. We first propose a technique, that clusters related success and error modes in a semantic embedding space. We show how VLMs can accurately identify the common trend in the error modes by contrasting these success and error modes, and propose a technique to select data based on the identified effects. Our experiments show that we improve performance across large datasets in an interpretable manner.

## 2 RELATED WORK

In order to identify a suitable approach towards solving the above mentioned problems, the related work is analyzed. This analysis focuses on three key aspects. First, clarifying what constitutes an insufficiency and how it can be measured even if no ground truth is known. Secondly, how to identify and isolate an insufficiency for a model. Third, having identified an insufficiency in a model, how can a correlating explanation or "effect" be extracted that accurately describes the insufficiency.

In this work, we focus on performance insufficiencies arising from deep learning-based object detectors. Performance insufficiencies are typically evaluated as pairwise distances, as in Caesar et al. (2020), or volume intersection between predicted and ground truth objects as in Geiger et al. (2012). Other methods, such as proposed by Sharma et al. (2021) and Majumdar et al. (2025), estimate potential insufficiencies by detecting out-of-distribution (OOD) samples.

To uncover systematic error sources in AI-based systems, it is necessary to identify subgroups within the dataset where performance insufficiencies occur. Addressing such insufficiencies requires locating Region of Interests (ROIs), regions that contain data points sharing similar characteristics, and common error causes. Approaches proposed by d'Eon et al. (2022), Eyuboglu et al. (2022), and Jain et al. (2023) cluster embeddings to find error-prone subgroups, sometimes extending into cross-modal spaces and generating textual descriptions. However, these approaches mostly focus on where insufficiencies occur without uncovering the nuanced underlying cause of insufficiency. Approaches like ADA-Vision (Gao et al., 2023) or retrieval-based methods (Rigoll et al., 2023) further introduce human-in-the-loop refinement or data collection, but at the cost of automation or interpretability.

After identifying ROIs that contain equivalent insufficiencies of the AI model, the next step is to provide a precise characterization of the samples within these regions. Li et al. (2023) proposes a mechanism to decode image embeddings into textual descriptions. While useful for caption generation, its primary goal is descriptive fidelity rather than diagnosing factors correlating with model errors. VisDiff (Dunlap et al., 2024) and the framework proposed by Zhong et al. (2023) employ language models to articulate differences between subsets of data. Based on that Greer & Trivedi (2024) present an interpretable novelty detection approach by identifying out-of-distribution samples and providing explanations about the novelty. In follow-up work (Greer et al., 2025), they demonstrate the effectiveness of their proposed method for active learning in 3D object detection. Nevertheless, this approach is limited in terms of the identification and characterisation of out-of-distribution or rare samples, while it does not capture model insufficiencies.

Across these directions, two main limitations emerge. First, prior work often separates detection and explanation, either identifying insufficiencies without explaining them or describing differences in the dataset without linking them to model errors. Second, methods frequently rely on either strong annotation requirements or human involvement, limiting scalability.

Figure 1: Architecture of the 1st stage. • symbolizes each object embedding $e_{o,i}$ of object $s_{o,i}$ within the embedding space, the detection performance is displayed by the color of the symbol. ★ and ★ denote the positive and negative anchor embeddings for ROI identification. Positive and negative subsets of the ROI $\mathcal{D}_\mathcal{R}^+$ and $\mathcal{D}_\mathcal{R}^-$ are then fed to the VLM to extract common concepts $\varepsilon_C$ and differentiating concepts $\varepsilon_D$.

## 3 PROBLEM FORMULATION

The objective of this work is 1) to identify patterns in the error modes of vision-based object-detection algorithms for automated driving, and 2) to subsequently leverage such insights towards model improvement. Concretely, let $\mathcal{M}_{baseline}$ be a pre-trained perception model, for which we assume we do not have access to the original training dataset. To identify issues with the base model $\mathcal{M}_{baseline}$ we assume access to a *labeled* validation dataset $\mathcal{D}_{val}$ and a performance metric $p$ so that we can quantify the model's performance on a large set of samples. While doing so is standard, basic practice to measure performance and compare models, in this work we further use $\mathcal{D}_{val}$ to pinpoint specific, targeted error modes of the base model. That is, we firstly aim to discover semantically coherent "effects" that correlate with model degradation, which we can then use to improve model performance in an interpretable manner.

Specifically, we use the identified semantic effects to guide additional data collection. We consider a setting where we have a large pool of unlabeled data $\mathcal{D}$, from which we may select a limited subset $\mathcal{D}_{select}$ with $|\mathcal{D}_{select}| \ll |\mathcal{D}|$ for labeling. This is typical for autonomous driving, as AVs generate vast amounts of data during deployment, not all of which can be labeled under budget constraints. Therefore, we must select samples that most align with the identified performance-degrading effects, so that retraining on $\mathcal{D}_{train} \cup \mathcal{D}_{select}$ incrementally addresses the model's error modes.

## 4 STAGE ONE: EFFECT EXTRACTION

The first stage is designed to identify and describe the insufficiencies of a model. An overview of the architecture is depicted in figure 1. In order to identify insufficiencies, the preexisting training dataset is processed and augmented with additional information. In order to focus the later analysis on the objects to detect, each ground truth object $s_{o,i}$ within each sample of the dataset is identified and extracted. Following each cropped object is embedded using the embedding model $E(\ )$, in this work we use CLIP ViT-L-14(Radford et al., 2021), yielding $e_{o,i}$, a compact representation that captures the semantic properties of the image. To quantify insufficiency, the Intersection over Union (IoU) of each ground truth object is calculated for the best fitting object predicted by the baseline model $\mathcal{M}_{baseline}$. To isolate relevant ROIs, we propose a Monte Carlo sampling approach guided by post-hoc evaluation criteria. This method generates candidate ROIs in a semantic embedding space by sampling candidate regions in a semantic embedding space and subsequently verifying their suitability. The presented method is designed with an generative approach in mind, focusing on an anchor embedding with poor performance from which one individual ROI is going to be built. We do this because, although a basic clustering algorithm like K-means can effectively cluster a dataset based on semantics, we want the region of interest to center on the model's error modes. The generation process for a ROI consists of the following primary steps:

First, the anchor samples, used to initialize the subregions of the ROI, without (positive) and without (negative) insufficiencies are defined. The negative anchor $e_\star^-$ is randomly chosen, with the proba-

Figure 2: Architecture of the 2nd stage. ■ denote the embeddings $e_k$ of samples $s_k$, while ★ denotes the anchor embedding $e_{\star,\text{train}}$

bility inversely weighted by the sample performance, from the embedding space. The ROI is defined by selecting all samples above a similarity threshold $sim_{\text{crit}}$. Following, the positive anchor $e_\star^+$ is identified based on the following criteria:

1. the performance difference must exceed a minimum threshold

$$p(e_\star^+) > p(e_\star^-) + \Delta p_{\min}$$

2. the cosine-similarity must be above a threshold

$$sim(e_\star^-, e_{\text{o},i}) > sim_{\text{crit}} \quad \text{with} \quad sim(e_i, e_j) := \frac{e_i \cdot e_j}{\|e_i\|\|e_j\|}$$

3. the performance gradient must be maximal

$$e_\star^+ = \arg\max_{e_{\text{o},i}} \frac{p(e_\star^-) - p(e_{\text{o},i})}{sim(e_\star^-, e_{\text{o},i})}$$

Having identified the anchors, the identified ROI $\mathcal{D}_{\mathcal{R}}$ is divided into positive and negative candidate subsets, $\mathcal{D}_{\mathcal{R},\text{cand}}^+$ and $\mathcal{D}_{\mathcal{R},\text{cand}}^-$ respectively, based on a performance threshold.

$$e_{\text{o},i} \in \begin{cases} \mathcal{D}_{\mathcal{R},\text{cand}}^+ : p(e_{\text{o},i}) \geq p_{\text{crit}} \\ \mathcal{D}_{\mathcal{R},\text{cand}}^- : p(e_{\text{o},i}) < p_{\text{crit}} \end{cases} \quad \text{with} \quad p_{\text{crit}} = \frac{p(e_\star^+) + p(e_\star^-)}{2}$$

In order to distill each subset down to a suitable magnitude for further processing, we employ the `Farthest Point Sampling` algorithm. `Farthest Point Sampling` iteratively adds samples that are maximally separated from the already selected dataset, thereby preserving diversity of samples (see Appendix B). Thus yielding $\mathcal{D}_{\mathcal{R}}^+$ and $\mathcal{D}_{\mathcal{R}}^-$. The algorithm is chosen to ensure good coverage of the embedding space within the ROI and to prevent an over-representation of too similar samples in the subset. This effect can be observed, for example, when the recording vehicle is stationary at an intersection.

A VLM is utilized to extract differentiating effects between the ROI subsets. The input consists of the two sets of samples from the first stage, $\mathcal{D}_{\mathcal{R}}^+$ and $\mathcal{D}_{\mathcal{R}}^-$, one associated with high performance and another with low performance. Each set includes 100 samples, chosen to balance representativeness and VLM context length constraints. The model is prompted to identify both differentiating attributes $\varepsilon_{\text{D}}$, that explain the performance gap and common attributes $\varepsilon_{\text{C}}$, that are shared by both subsets. These text based effect descriptions serve as semantic indicators of data insufficiencies and are used to guide the sampling of new data. For these experiments `gemini-2.0-flash` was chosen as a VLM model.

## 5 STAGE TWO: MODEL IMPROVEMENT

The second stage is aimed at leveraging the effects identified by the first stage in order to improve the performance of the baseline model. An overview of the architecture is depicted in figure 2. Addressing insufficiencies identified through the VLM analysis requires realistic data that exhibit these

identified effects. A viable strategy is targeted recording, in which additional real-world data are either deliberately collected or sourced from existing datasets to ensure the relevant insufficiencies are represented.

Taking these considerations and the scope of this paper into account, the first approach, targeted recording of additional real-world data, is chosen. We construct an anchor embedding $e_{\star,\text{train}}$ for the training sample selection by adding the common effect embedding $e_{\text{C}}$ to the differentiating effect embedding $e_{\text{D}}$.

$$e_{\star,\text{train}} = E(\varepsilon_{\text{C}}) + E(\varepsilon_{\text{D}})$$

To build the training set, we extract samples that are similar to this anchor, as we assume that the training set and anchor embedding can be from different datasets. At the same time, we also want to introduce diversity for this to be trained effectively. As such, we filter all candidates for a threshold similarity towards the anchor and then try to achieve good sampling coverage in the subspace by employing `Farthest Point Sampling` for $k + 1$ samples, to account for the virtual anchor used for algorithm initialization. This approach to data provisioning enables the transfer of the identified effects from a preexisting dataset to any new dataset by using the semantic effect descriptions. Further, it also enables the identification of effects on object level, while the training data can be drawn at the original object level (here images).

Besides targeted data recording, other promising strategies include data generation through full synthesis, which involves constructing entirely artificial scenes from scratch while carefully embedding the identified insufficiencies, and data augmentation, where existing realistic scenes are systematically modified or perturbed to incorporate these insufficiencies and create additional scenario variants. Both strategies hold potential for future work, yet they share the central challenge of realistically capturing the complexity of the real world.

## 6 EXPERIMENTS

In order to evaluate SPIDER the performance of the fine-tuned model is compared against the performance of the baseline model. As a reference, a naive approach of randomly sampling an equally sized dataset for fine-tuning is implemented. For the experiments `Yolo11n` (Khanam & Hussain, 2024) pretrained on the COCO dataset (Lin et al., 2014) is used as the baseline model, due to considerations of training time. For model improvement, the NuScenes dataset (Caesar et al., 2020) is employed. Mean Average Precision with IoU=[0.5:0.05:0.95] ($\text{mAP}_{50-95}$) is used as established metric for model improvement. All validation of the models is performed on the NuScenes dataset. Two basic experiments are set up to evaluate SPIDER .

### 6.1 SINGLE-ROI EXPERIMENT

This experiment aims at evaluating the effectiveness of the first stage of SPIDER . For this, the following Hypothesis will be tested.

**Hypothesis**: *The presented approach can effectively identify insufficiencies in the perception model.*

In order to quantify the effectiveness of the first stage, a single ROI is identified and used for model improvement in the second stage. We evaluate end-to-end, since it is not possible to ablate whether the clustering yields non-trivial points for improvement. The resulting increase in $\text{mAP}_{50-95}$ of the model correlates with the amount of novel, and therefore previously insufficient, information within the dataset. This increase is compared to the increase caused by using an equivalently sized, but randomly sampled, set of images. Thus we can quantify the ratio of additional novel information extracted by applying SPIDER . In order to account for stochastic influence of the `Farthest Point Sampling` algorithm in the second stage, each identified ROI is independently evaluated five times.

### 6.2 MULTI-ROI EXPERIMENT

Building on the previous experiment, the question arises as to how strongly SPIDER can be leveraged. To qualify this, we aim to answer two key questions:

1. How many samples are sufficient for each ROI?

2. Can an upper bound for the number of ROIs be determined?

In this experiment, multiple ROIs are identified, for each ROI a number of samples are collected. For the reference approach the number of used samples is equal to the sum of samples for all ROIs. We vary both the number of ROIs and the number of samples per effect.

## 7  RESULTS

In this section, we examine the results of the performed experiments. Each experiment was performed on a single H100 GPU.

Figure 3 shows the results of the single-ROI experiment. Overall the $\mathrm{mAP}_{50-95}$ improvement for each ROI is larger than the maximum improvement from the reference approach. While the reference approach varies between an $\mathrm{mAP}_{50-95}$ of 0.117 and 0.123 the oberserved ROIs vary between 0.125 and 0.141. On average SPIDER outperformed the reference approach by 10% to 14%. We conclude that hypothesis 1 holds, and SPIDER can effectively identify insufficiencies in the perception model.

Table 1 expands on the ROIs with more details. Each ROI can be considered to represent a different quality. While $\mathcal{R}_1$ focuses on object type, $\mathcal{R}_2$ focuses on environmental conditions and $\mathcal{R}_3$ on image quality. While the similarity of the anchors $sim(e_\star^+, e_\star^-)$ is quite high, the mean performance values for the positive and negative subsets vary strongly, highlighting that the positive and negative categories are only relative and not bound to fixed threshold. Overall the ROIs showcase, that SPIDER is able to capture subtle differences between positive and negative subsets.

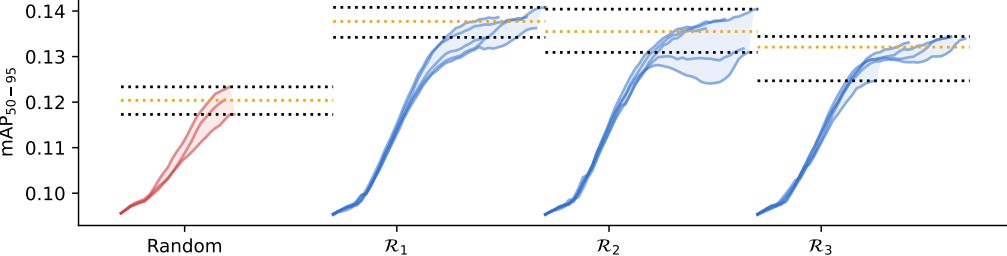

Figure 3: $\mathrm{mAP}_{50-95}$ development for the reference random sampling approach and different ROIs $\mathcal{R}_i$. ⋯⋯ denotes the maximum and minimum best $\mathrm{mAP}_{50-95}$ value, while ⋯⋯ denotes the mean result. Details for the ROIs are shown in Table 1

The results of the multi-ROI experiment are shown in Figure 4. For the sampling settings of 10 and 25 samples per effect, SPIDER consistently outperforms the reference baseline of randomly selected data. As illustrated in Figure 4, we observe a consistent increase in $\mathrm{mAP}_{50-95}$ for SPIDER , with improvements of approximately 13% over the reference for these configurations. Using the sampling setting of 50 samples per effect, the performance difference between SPIDER and the reference approach is negligible. This shows that there is a critical number of samples per ROI, which limits the effectiveness that can be used.

Further, the results show that increasing the number of distinct effects used for model fine-tuning improves $\mathrm{mAP}_{50-95}$. However, this improvement exhibits diminishing returns as the number of effects grows. This phenomenon is likely due to saturation of novel insufficiency-related information within the used small scale dataset. Once the major insufficiencies are addressed, newly discovered effects may represent less impactful or overlapping error modes, reducing the marginal benefit of additional fine-tuning.

## 8  ABLATION STUDY

To evaluate the effectiveness of VLMs in identifying insufficiency correlated effects in perception-related tasks, we design an ablation study that assesses the VLMs ability to semantically distinguish between two sets of images. The designed ablation study, depicted in Figure 5, is intended to

Table 1: Details of the ROIs displayed in Figure 3. For each ROI one image sample of the positive and negative subset, the common $\varepsilon_C$ and differentiating $\varepsilon_D$ effects, the mean performance of the positive ($\bar{p}_{\mathcal{R}}^+$) and negative ($\bar{p}_{\mathcal{R}}^-$) subset, as well as the similarity between the positive and negative anchor is shown.

| | $\mathcal{R}_1$ | $\mathcal{R}_2$ | $\mathcal{R}_3$ |
|---|---|---|---|
| $s_{o,i} \in \mathcal{D}_{\mathcal{R}}^+$ | | | |
| $s_{o,i} \in \mathcal{D}_{\mathcal{R}}^-$ | | | |
| $\varepsilon_C$ | Vehicles Low Image Quality Blur Roads/Parking Areas | People (often viewed from the back) Urban environment (streets, sidewalks) Low Image Quality (blurriness, noise) | People Outdoor environment Graininess |
| $\varepsilon_D$ | Prominent Box Trucks Clearer Vehicle Details | Overcast or Rainy Weather Conditions Presence of Cars Road Markings or Construction Barriers | Night time setting Low light conditions Reflective materials |
| $\bar{p}_{\mathcal{R}}^+$ | 0.91 | 0.64 | 0.60 |
| $\bar{p}_{\mathcal{R}}^-$ | 0.19 | 0.22 | 0.06 |
| $sim(e_\star^+, e_\star^-)$ | 0.90 | 0.91 | 0.95 |

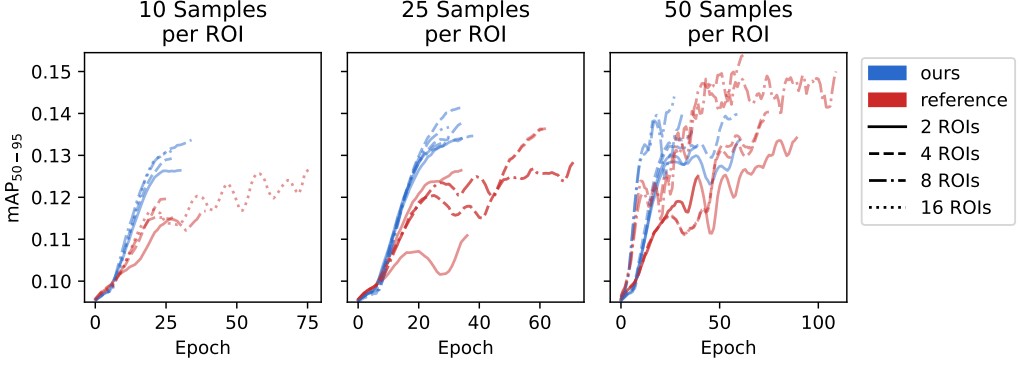

Figure 4: Continuous application of first and second stage over 20 iterations. The size of samples used from each ROI is varied between 50 and 400 samples. (Top): Model improvement compared to the baseline model. (Bottom): Advantage of SPIDER , relative to the reference approach.

demonstrate the method for larger scale datasets, where a higher quantity and variance in suitable metadata is given. For the study, we partition cropped images from the NuScenes dataset based on perception-relevant concepts derived from the annotated data. To quantitatively determine whether the concept identified by the VLM aligns with the true underlying discriminative factor, we em-

Figure 5: Methodology for ablation study and LLM-as-a-Judge calibration.

ploy a Large Language Model (LLM) to rate the semantic similarity between the predicted and ground-truth concepts in the context of environmental perception. To validate the reliability of this automated evaluation, we perform an additional ablation study comparing the VLM-generated similarity ratings with human-provided judgments. Overall, we conduct an ablation study to rate the LLM's capability as a semantic similarity judge and to rate the VLM's capability to identify the key differentiating concept between two image sets.

## 8.1 LLM-AS-A-JUDGE CALIBRATION

To assess whether the identified concept aligns with the true concept, semantically for environmental perception, the LLM is instructed to provide a rating between 1 to 4. A rating of 1 indicates that the concepts are unrelated, while a rating of 4 denotes semantic equivalence. The specific interpretation of each rating level is predefined and included in the prompt (see Appendix C).

For the ablation study, we define a tuple of scene-relevant concepts $\varepsilon_1$ and $\varepsilon_2$, related to perception tasks in automated driving, such as objects and surrounding environment. We then assign concept pairs to each discrete rating category. In total, we evaluate 48 concept pairs, approximately evenly distributed across all rating categories. Each pair is rated four times by the LLM to assess the consistency of the response. The average variance of across these four responses for each pair is $0.04$, indicating small deviations in the LLM's judgments. Table 2 compares the LLM ratings with our annotated ground truth. These results suggest that the LLM ratings are closely aligned with our judgements. A linear regression fitted to the data yields a correlation coefficient of $0.84$. It is important to note that assigning semantic similarity scores inherently involves subjective interpretation. Consequently, the annotated ground-truth ratings are not strictly objective. The observed minor deviations between the LLM and ground-truth rating can also be attributed to this subjectivity. Overall, the results indicate that the LLM is capable of providing reliable and consistent ratings of semantic equivalence between concept pairs, supporting its use as a judge for large scale data sets.

## 8.2 EFFECT EXTRACTION

This ablation study investigates whether the VLM can identify distinguishing concepts between different sets of images to enable it to later detect error-influencing factors by comparing image sets with and without perception errors. To this end, we construct image set pairs by categorizing cropped NuScenes images according to available annotations. In each experiment, the VLM is presented with two image sets and prompted to identify the most important concept present in one set but absent in the other. Both environment-specific pairs (*day vs night, rain vs no rain*) and object-specific pairs (*sitting/lying pedestrian vs moving pedestrian, adult vs child, police vs bicycle, car vs truck, motorcycle vs bicycle, car vs pedestrian, police vs adult*) are examined.

To evaluate whether the concept identified by the VLM aligns with the original annotation used to construct the image sets, we apply a secondary prompt where the LLM acts as a judge and assigns a rating from 1 to 4, as previously described. Each pair of images is tested in ten trials. In each trial, 20 cropped images are randomly sampled for each concept from the complete NuScenes dataset and provided to the VLM. Table 3 illustrates the results. The identified concepts mostly align with annotations of the image set, as reflected by the dominance of ratings 4 and 3. However, in some cases, the rating is 1, indicating no semantic equivalence. In these cases, the VLM either fails to identify the intended effect or instead captures a different, seemingly more dominant factor that distinguishes the image sets.

Table 2: Evaluation data for using LLM-as-a-judge. Here the LLM ratings are compared against the human generated ground-truth ratings.

|  | | | Ground Truth Rating | | |
|---|---|---|---|---|---|
|  | | **1** | **2** | **3** | **4** |
| | **1** | 37 | - | - | - |
| **LLM Rating** | **2** | 12 | 21 | - | - |
| | **3** | 1 | 29 | 68 | 40 |
| | **4** | - | - | 2 | 30 |

Table 3: Effect extraction quality of the VLM as rated by LLM-as-a-judge

| Rating | 1 | 2 | 3 | 4 |
|---|---|---|---|---|
| Count | 11 | 4 | 14 | 41 |

## 9 LIMITATIONS

This study presents several limitations worth noting. First, the proposed two-stage approach heavily relies on VLMs to extract causally correlated factors and generate semantic descriptions of data insufficiencies. The accuracy and effectiveness of these VLM-generated insights are inherently dependent on the characteristics of the VLM employed, introducing potential biases or inaccuracies that might propagate into later stages of data provisioning and model fine-tuning. Another limitation of the first stage of SPIDER is, that while ideally the system would be able to identify effects causally related to the insufficiency, it can only identify effects correlated to the insufficiency. That is, if the model struggles to correctly detect people holding umbrellas, it is quite possible, that "rain" might be identified as the effect instead. Moreover, the use of targeted real-world recordings for addressing identified insufficiencies, although effective, may introduce logistical and economic challenges. Such targeted data collection efforts can be resource-intensive, potentially limiting the scalability and practicality of this method for extensive, diverse applications. While other approaches to data provisioning, such as scene generation and augmentation, exist, fully exploring their effects is beyond the scope of this paper and thus remains to be validated. Further, the used `Yolo11n` object detection model may not be representative of more complex and capable perception models used for automated driving systems. While the model is suitable for showing an initial proof of concept, validation on more complex perception systems is needed.

## 10 CONCLUSION

This research proposes SPIDER to systematically address and improve insufficiencies within automated driving perception systems. By leveraging VLMs, this novel two-stage approach effectively identifies and characterizes model insufficiencies through semantic analysis, subsequently guiding targeted data provisioning to mitigate these deficiencies. The experimental evaluations shows effictiveness of SPIDER , demonstrating advantages in model improvement. Specifically, SPIDER showed consistent performance improvements across both single and multi-effect scenarios, highlighting its potential applicability in model improvement settings. Moreover, identifying factors contributing to insufficiencies in the first stage enhances the understanding of model limitations.

SPIDER is particularly useful for large-scale datasets. Manually scanning millions of samples to uncover subtle, context-dependent insufficiencies is prohibitively time consuming. Human experts would often need many hours to detect patterns that SPIDER identifies automatically from semantically coherent ROIs in the embedding space. By contrasting nearby high- and low-performance samples rather than relying on a single absolute threshold, SPIDER remains effective even when such thresholds are unavailable, ambiguous, or disputed across tasks and metrics. This design enables practical large-scale use while preserving fine-grained, interpretable error mode descriptions.

Besides model improvement, the knowledge can also be used to restrict the OD of automated driving system deployment or develop fusion strategies with other perception algorithms based on the gained insights about the identified model limitations. Further research is recommended to explore more sophisticated continual learning techniques and validate SPIDER across broader datasets, data provisioning strategies and more complex perception systems. This work provides a step towards enhancing the transparency and explainability of perception models, contributing positively toward safer and more reliable automated driving systems.

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

# A  PROMPT

---

Listing 1: Prompt used for first stage VLM

---

```
 1  You are tasked with comparing two sets of images ("Set 1" and "Set 2")
        .
 2  Both sets of images contain objects that a computer vision model was
        tasked to detect.
 3  In one set this task was significantly better performed than the other
        .
 4  You job is to identify the most differentiating concepts between the
        two sets.
 5  For this, you have to analyze the visual content, themes, and
        characteristics of each set, and then determine what key elements
        distinguish one set from the other.
 6  These concepts can concern the object within the picture, the overall
        image composition or any other relevant visual elements.
 7  Examples are object type, color, size, position, environment
        conditions, weather conditions, brightness, blur, focus, etc.
 8
 9  Please carefully examine both sets of images and determine the key
        common concepts that best describe the concepts present in each
        individual image.
10  Please carefully examine both sets of images and determine the key
        concept that best differentiate the two sets from each other.
11
12  Aim to qualify the key concepts with as much detail as possible.
13  For example, instead of saying "color", specify the color or color
        palette that is most significant.
14  Instead of saying "high quality," describe the specific attributes
        that make the quality high.
15  Instead of saying "different object types", specify how the objects
        differ, e.g. "elderly pedestrians" vs. "children".
16  Each concept should represent a unique aspect that distinguishes one
        set from the other.
17  Remember to base your analysis solely on the visual information
        provided in the two sets of images. Do not make assumptions about
        information that is not visually present.
18
19  <images Set="Set_1"></images>
20  <images Set="Set_2"></images>
21
22  Your complete response should be structured as follows:
23      <analysis>
24      [Your detailed analysis and reasoning here]
25      </analysis>
26
27  Your complete response must be structured as follows:
28      <common_concepts>
29       [Most significant common concept]
30       [Second most significant common concept]
31       [Third most significant common concept]
32       [Additional concepts if necessary]
33      </common_concepts>
34      <differentiating_concepts present_in="Set_1">
35       [Most significant differentiating concept in maximum 2 Words (
            preferably one), please focus on one distinguishable concept]
36      </differentiating_concepts>
37      <differentiating_concepts present_in="Set_2">
38       [Most significant differentiating concept in maximum 2 Words (
            preferably one), please focus on one distinguishable concept]
39      </differentiating_concepts>
```

---

## B    FARTHEST POINT SAMPLING

The following pseudocode describes the general function of the `Farthest Point Sampling` algorithm. Within the proposed approach the anchors $e_\star$ are used as initial points. In the second stage, where the anchor is a virtual point, $k$ is increased by one and the anchor is later dropped from the resulting point list. As the algorithm is used in the embedding space, cosine distance is used as a distance metric.

$$d_{\text{cosine}}(\mathbf{e_i}, \mathbf{e_j}) = 1 - \frac{\mathbf{e_i} \cdot \mathbf{e_j}}{\|\mathbf{e_i}\|\|\mathbf{e_j}\|}$$

---

Algorithm 1: Farthest Point Sampling (FPS)

---

**Require:** Point set $P = \{p_1, p_2, \ldots, p_n\}$, sampling size $k$
**Ensure:** Sampled subset $S \subseteq P$, $|S| = k$
 1: Initialize empty set $S$
 2: Select initial point $s_1 \in P$ arbitrarily (or using a heuristic)
 3: $S \leftarrow \{s_1\}$
 4: **for** $i = 2$ **to** $k$ **do**
 5:     $maxDist \leftarrow -\infty$
 6:     $nextPoint \leftarrow$ null
 7:     **for all** $p \in P \setminus S$ **do**
 8:         $dist_p \leftarrow \min_{s \in S} d(p, s)$
 9:         **if** $dist_p > maxDist$ **then**
10:             $maxDist \leftarrow dist_p$
11:             $nextPoint \leftarrow p$
12:         **end if**
13:     **end for**
14:     $S \leftarrow S \cup \{nextPoint\}$
15: **end for**
16: **return** $S$

---

## C  JUDGE PROMPT

Listing 2: Prompt for LLM-as-a-judge

```
 1  You are asked to evaluate whether two concepts, a ground truth and a
        prediction, have the same semantic meaning. These concepts
        represent common error scenarios/causes in automated driving
        vision systems.
 2  Use the following rating to evaluate each concept pair:
 3  Name  Rating  Description
 4  Not Helpful  1  The match between both concepts is terrible:
        completely irrelevant
 5  Mostly_not_helpful  2  The match between both concepts is mostly not
        helpful: mismatch in some key aspects and details
 6  Mostly_helpful  3  The match between both concepts is mostly helpful:
        matches in most key aspects and details
 7  Helpful  4  The match between both concepts is excellent: both
        concepts are semantically equivalent
 8
 9  Use these examples as references:
10  ID  Concept of Interest Predicted Concept Rating
11  B1  Pedestrian  Child 4
12  B2  Urban Environment Buildings 3
13  B3  Stop Sign Traffic Jam 2
14  B4  Intersection  Occlusion 1
15
16  Task:
17  <concept_of_interest></concept_of_interest>
18  <predicted_concept></predicted_concept>
19  Your complete response should be structured as follows:
20  <analysis>
21  [Your detailed analysis and reasoning here]
22  </analysis>
23  Differentiating Concepts present in "Set 1", but not in "Set 2":
24  <rating>
25   [Your rating name here]-[Your numerical rating here]
26  </rating>
```

