# OpenReview forum: "Uncovering And Correcting Perception Model Weaknesses Using VLM-Based Analysis"
_ICLR.cc/2026/Conference — Submitted to ICLR 2026_

### Official Review · Reviewer_bK4m · 2025-10-25

**Soundness:** 2
**Presentation:** 1
**Contribution:** 1
**Rating:** 0
**Confidence:** 4

**Summary:**

The paper tries to identify and correct perception model weaknesses via vision–language model (VLM) analysis. The high-level goal of integrating explainability and model improvement into a unified pipeline is relevant.
However, the paper is methodologically unsound and lacks both novelty and rigorous experimental validation. Its claims are unsubstantiated, and the writing quality is far below the standard for ICLR.

**Strengths:**

There are no discernible strengths in this submission. The idea of using VLMs to find model failure modes is reasonable, but the execution is so profoundly deficient that it completely nullifies any potential merit.

**Weaknesses:**

1. Severely flawed experimental design and validation. The paper validates SPIDER only on a single dataset (nuScenes) and with a weak baseline detector (YOLO11n pre-trained on COCO). This model is not representative of advanced perception systems for autonomous driving. Therefore, the “discovered weaknesses” likely reflect domain mismatch rather than genuine model insufficiencies. The experiments do not demonstrate that SPIDER can uncover subtle or previously unknown failure modes in strong models. Consequently, the core claim that SPIDER identifies “hard-to-detect insufficiencies in perception models” is unsupported.
2. Methodological opacity and incoherence. The paper’s two-stage pipeline is poorly described.
- Section 4 is confusing and inconsistent: it never clearly explains how the object crops are obtained (e.g., from ground-truth boxes, detector outputs, or heuristic proposals), how embeddings are computed (what encoder or feature layer is used, with what dimensionality), or what (e_{o,i}) specifically represents (a token embedding, pooled feature, or latent descriptor). These missing details make it impossible to reconstruct or reason about the pipeline.
- The “anchor” mechanism is undefined (dimensionality, derivation, and relation to embeddings are all missing).
- Stage 2 (“Correction”) is incomprehensible — there is no clear explanation of how semantic effects are selected or used to retrain models.
3. Incoherent writing and formatting issues. Numerous grammatical errors, inconsistent notation, unclosed parentheses, and malformed sentences (e.g., “without and without insufficiencies”). Section 4’s mathematical symbols are undefined. Figures and legends (notably Fig. 4) do not correspond. The manuscript appears unedited and unproofread, making comprehension extremely difficult.
4. Unsupported claims and lack of alignment with introduction. The introduction claims that SPIDER integrates detection, explanation, and remediation (e.g., Line 61), but in reality, all three steps remain decoupled. The experiments fail to demonstrate any integrated correction loop. There is no evidence that the system meaningfully improves model robustness.
5. Insufficient experimental depth.
- Only a single dataset and model are tested — no cross-domain validation or scalability analysis.
- No comparison with baselines or established explainability methods (e.g., ADA-Vision, UnSAM).
- Computational cost and scalability of the VLM-based analysis are unreported, leaving practicality unclear.
6. Terms like “semantic effects” are introduced without definition. It remains unclear whether these represent text embeddings, visual clusters, or manually curated labels. The pipeline’s terminology (anchor, subregion, effect) shifts meaning across sections, leading to further ambiguity. The Related Work section also contains several unclear or logically inconsistent claims: for example, Line 93 (“…at the cost of automation or interpretability”) is logically confusing — it is unclear why the mentioned approaches would necessarily reduce interpretability; Line 103 (“…does not capture model insufficiencies”) contradicts its own premise since identifying rare or out-of-distribution samples can itself be viewed as capturing insufficiencies; and Line 127 (“patterns in the error modes”) is redundant and awkwardly phrased, as both ‘patterns’ and ‘modes’ describe forms of structure — consider simplifying to “types of errors” or “systematic failure cases.” These issues indicate that key concepts and relationships are not clearly articulated, which further undermines the interpretability and coherence of the paper.
7. Overall lack of scientific contribution. The proposed method appears to be a loosely assembled workflow combining standard VLM embedding, k-means clustering, and model retraining, without methodological novelty or insight. It reads more like an engineer demo than a research paper.

**Questions:**

See more and main in the weakness section. Here are other questions.
1. The paper only evaluates SPIDER on YOLOv11 trained on COCO, which is a general 2D detector and not representative of perception systems used in autonomous driving. Why not test with a stronger in-domain 2D detector trained on driving datasets (e.g., DETR, DINO-DETR, or Mask R-CNN on nuScenes/BDD100K)? If SPIDER were applied to such models, would it still uncover non-trivial or previously unseen failure modes, or would it simply reflect dataset bias?
2. The effect extraction stage depends on the responses of a Vision-Language Model when contrasting two image sets. However, the paper provides no analysis of this process’s stability or reproducibility. For the same pair of image sets, does the VLM consistently produce the same “semantic effects,” or do its outputs vary across runs due to stochastic decoding or prompt sensitivity? Without such an evaluation, it is unclear whether SPIDER can reliably identify the same failure factors across repeated analyses or across different VLMs.
3. The abstract broadly claims that SPIDER addresses perception systems in general, yet all experiments are limited to 2D object detection. If the method has only been tested in this context, the claims should be narrowed accordingly. Alternatively, if SPIDER is intended to generalize, please demonstrate or discuss its applicability to other perception tasks such as semantic segmentation or tracking, clarifying which components are task-agnostic and which are specific to object detection.
4. The paper claims that Stage 2 “corrects” the model based on the identified semantic effects, but this process is never specified. It remains unclear whether the model is actually retrained or merely re-evaluated, what data selection strategy is used, and what loss function or optimization objective governs this correction. Without these details, the so-called remediation stage is conceptually vague and experimentally unverifiable.

---

### Official Review · Reviewer_iako · 2025-10-29

**Soundness:** 2
**Presentation:** 3
**Contribution:** 2
**Rating:** 4
**Confidence:** 3

**Summary:**

This manuscript proposes SPIDER, a novel two-stage method for diagnosing and addressing insufficiencies in automated driving perception models using VLMs. In the first stage, the method segments data in a semantic embedding space to identify  ROI with divergent detection performance, then leverages VLMs to extract interpretable semantic effects correlated with model errors. The second stage uses these effects to guide targeted data selection and model fine-tuning to mitigate the identified weaknesses. Evaluations on the NuScenes dataset, with YOLO11n as the baseline model, demonstrate that SPIDER outperforms random sampling in improving model. This method enhances transparency and explainability of perception systems critical for automated driving safety.

**Strengths:**

SPIDER integrates error identification, semantic explanation, and targeted model improvement into a pipeline, addressing the decoupling of detection and improving in existing methods.

By using VLMs to extract human-understandable semantic effects from high and low-performance data subsets, the method overcomes the black-box limitation of traditional perception models, facilitating actionable insights for developers.

**Weaknesses:**

The accuracy of semantic effect extraction is highly reliant on the capabilities of VLMs and detection models, introducing potential biases or inaccuracies that propagate to data selection and model fine-tuning stages. No comparison across different VLMs or different detection models is provided to demonstrate robustness.

The framework only identifies factors correlated with model insufficiencies, not causal relationships. This may lead to misattribution (e.g., identifying "rain" instead of "umbrellas held by pedestrians" as the root cause of detection failures), limiting precise error mitigation.

Targeted real-world data collection for addressing identified insufficiencies may still face logistical and economic barriers. Alternative strategies like synthetic data generation or augmentation are not discussed in depth.

**Questions:**

How can SPIDER be enhanced to distinguish between correlated and causal factors underlying model insufficiencies?

What are the potential advantages and limitations of using synthetic data generation or data augmentation instead of targeted real-world data collection?

---

### Official Review · Reviewer_LVXo · 2025-11-10

**Soundness:** 2
**Presentation:** 2
**Contribution:** 2
**Rating:** 2
**Confidence:** 5

**Summary:**

This paper studies the out-of-distribution (OoD) detection and model correction problem in autonomous driving. The author proposes using the vision-language model (VLM) to identify the insufficiencies of an existing pretrained model while comparing the model output and the ground-truth, where the VLM is used to generate the semantic embedding on each ground-truth object and the best fitting objects predicted by the pretrained model. The insufficiency is measured by the IoU between the ground-truth and the existing pretrained model. The author then updates the model based on the identified insufficiency. Experiments are conducted on NuScenes datasets to verify the effectiveness of the proposed method.

**Strengths:**

1. The OoD detection and corresponding model correction are crucial to guarantee the safety-critical application like autonomous driving.

**Weaknesses:**

1. Missing relevant related works. Identifying the imperfection of a pretrained model and updating the model to correct the issue have been well studied, e.g., in [a, b, c, d], where these papers all considered leveraging vision-language models (VLMs) to help identify the issue and curate the dataset to update the model. Specifically, [c] proposed an automatic data engine that leverages VLMs and LLMs to identify the issue of the existing pretrained detector, curated a new training data tailored to the problem by unlabeled data, continually trains the existing detector, and finally verifies the updated model, all designed to be tailored to the Autonomous driving domain. However, all these papers are missing in the related work, and there is no comparison to these relevant works, making it hard to evaluate the significance of the proposed method.

2. Technical Novelty is limited and not well justified. First, in Stage One, the author utilizes the vision language model (e.g., CLIP) embedding for the cropped object to measure the IoU between the ground truth and the existing detector's prediction. However, we can also use any off-the-shelf detection model to get the embedding for the cropped image. The necessity of using VLM's embedding is not elaborated. Moreover, both Stage One and Two lack sufficient technical contributions, and it is challenging to convince the reviewer that the proposed design is significant.


3. Limited experimental analysis. In Section 7, no exisiting methods have been compared and only some simple baselines being compared. This is not sufficient to comprehensively verify the effectiveness of the proposed method.



Reference:

[a] Exploiting Unlabeled Data with Vision and Language Models for Object Detection. ECCV 2022

[b] Taming Self-Training for Open-Vocabulary Object Detection. CVPR 2024

[c] AIDE: An automatic data engine for object detection in autonomous driving. CVPR 2024

[d] Mcity Data Engine: Iterative Model Improvement Through Open-Vocabulary Data Selection. ITSC 2025

**Questions:**

Please refer to the Weaknesses section.

---

### Official Review · Reviewer_y8Gj · 2025-11-11

**Soundness:** 2
**Presentation:** 2
**Contribution:** 3
**Rating:** 4
**Confidence:** 3

**Summary:**

The paper introduces a two-stage framework that uses VLM to find and explain object detection model weaknesses by contrasting high- and low-performance samples in a semantic embedding space.
It then produces common and differentiating tags to guide targeted data selection and fine-tuning, improving model performance.

**Strengths:**

- An interesting approach to automatically identifying differences between high and low object detection performance for the same object type, providing human-understandable explanations.
- Good results are shown in the YOLO11n model using the Nuscenes dataset.

**Weaknesses:**

1. Although the paper claims that the proposed pipeline can scale automatically, the experiments do not clearly demonstrate this advantage. Only three ROIs in Table 1 are analyzed in detail, and Figure 4 shows experiments with at most 16 ROIs. It remains unclear how distinct these 16 ROIs are from each other, given that their characteristics are not described. As a result, the claimed scalability and automatic discovery of hard cases are not convincingly supported: if only a small number of ROIs are used, similar hard case tags could likely be manually marked or identified without the proposed approach.
2. Before generating object embeddings, the method crops the raw images into object sub-images using the labeled data position, if understood correctly. This implies that ground-truth annotations are required, yet the paper claims it selects subsets from unlabeled data (line 137). Please revise your statement if the claim is not true.
3. The paper only evaluates one setting, YOLO11n model on Nuscenes dataset.
4. It is unclear whether the VLM requires fine-tuning and how accurately it can generate common or distinguishing tags $\epsilon_{C}, \epsilon_{D}$ based on the positive and negative embedding set inputs.

**Questions:**

1. In line 184, why must the cosine similarity between e{−⋆} and e_{o,i} exceed a threshold? What is the purpose, considering e{−⋆} is randomly sampled?
2. The lines in Figure 4 are unclear, making it difficult to interpret or draw conclusions.

---

### Meta-Review · Area_Chair_WtPy · 2026-01-02

**Summary:**

All four reviewers tend to reject this paper, with the scores of 4,  2, 4, 0 (2.5 on average). I find the several major concerns from the reviewers:
(1) Weak performane. The experiments do not clearly demonstrate the advantages such as automatic scaling. Only one experimental setting is considered (YOLO11n on Nuscenes)
(2) Missing relevant related works. There are many works that aim to identify the imperfection of a pretrained model and update the model to correct the issue, such as the [a, b, c, d] rasied by reviewer LVXo.
(3) No comparison across different VLMs or different detection models is provided to demonstrate robustness.
(4) Incoherent writing and formatting issues. Numerous grammatical errors, inconsistent notation, unclosed parentheses, and malformed sentences (e.g., “without and without insufficiencies”). (Rasied by Reviewer bK4m)
(5) Unsupported claims and lack of alignment with introduction. The introduction claims that SPIDER integrates detection, explanation, and remediation (e.g., Line 61), but in reality, all three steps remain decoupled. The experiments fail to demonstrate any integrated correction loop. There is no evidence that the system meaningfully improves model robustness.

Overall, this paper falls below the acceptance criteria of ICLR such a top-tier conference.

**Reviewer Concerns:**

The authors did not provide rebuttal, and all concerns remain

**Reviewer Scores:**

All reviewers will keep scores and tend to reject. The final scores will be 4, 2, 4, 0 (2.50 on average).

---

### Decision · Program_Chairs · 2026-01-26

Reject